# Quantitative Analysis of Industrial Solid Waste Based on Terahertz Spectroscopy

Qingfang Wang, Qichao Wang, Zhangfan Yang, Xu Wu  and Yan Peng *

School of Optical-Electrical and Computer Engineering, University of Shanghai for Science and Technology, Shanghai 200093, China; 1935023504@st.usst.edu.cn (Q.W.); 1712440220@st.usst.edu.cn (Q.W.); 182390297@st.usst.edu.cn (Z.Y.); wuxu@usst.edu.cn (X.W.)
* Correspondence: py@usst.edu.cn

**Abstract:** Industrial solid waste refers to the solid waste that is produced in industrial production activities. Without correct treatment and let-off, industrial solid waste may cause environmental pollution due to a variety of pollutants and toxic substances that are contained in it. Conventional detection methods for identifying harmful substances are high performance liquid chromatography (HPLC) and gas chromatography-mass spectrometry (GC-MS), which are complicated, time-consuming, and highly demanding for the testing environment. Here, we propose a method for the quantitative analysis of harmful components in industrial solid waste by using terahertz (THz) spectroscopy combined with chemometrics. Pyrazinamide, benazepril, cefprozil, and bisphenol A are four usual hazardous components in industrial solid waste. By comparing with the Raman method, the THz method shows a much higher accuracy for their concentration analysis (90.3–99.8% vs. 11.7–86.9%). In addition, the quantitative analysis of mixtures was conducted, and the resulting prediction accuracy rate was above 95%. This work has high application value for the rapid, accurate, and low-cost detection of industrial solid waste.

**Keywords:** terahertz spectroscopy; industrial solid waste; quantitative analysis of mixtures



## 1. Introduction

Solid waste consists of urban solid waste, industrial solid waste, agricultural solid waste, and hazardous waste [1]. Some of the industrial solid wastes are toxic and dangerous. For example, pyrazinamide, benazepril, cefprozil, and bisphenol A are four usual hazardous components in industrial solid waste. Pyrazinamide is an effective medicine for tuberculosis, but it also has serious side effect, such as hepatotoxicity [2]. Benazepril is an effective medicine for high blood pressure, but the wrong dosage can lead to dizziness, cough, and other problems [3,4]. Cefprozil is used to treat respiratory and skin infections but may cause gastrointestinal adverse reactions [5]. Bisphenol A as a high production volume (HPV) chemical is commonly used in food packaging materials, dental sealants, medical devices, and thermal receipts, but it is also a reproductive toxicant in animal models [6]. If these wastes are not treated properly before discharge, they will have a serious impact on the environment and human beings, such as soil pollution, groundwater pollution, and pathogen breeding [7,8]. Therefore, these solid wastes must be strictly detected and treated.

Currently, for the toxicity judgment of industrial solid waste, the first step is to soak the waste, and then to detect the leaching solution after shaking and filtering [9]. Conventional detection methods include high performance liquid chromatography (HPLC) and gas chromatography-mass spectrometry (GC-MS), which have a high identification accuracy. However, these methods have the disadvantages of being time consuming, chemical reagent-using, and operation-cumbersome [10,11]. Moreover, these methods require a laboratory detection environment, which are unavailable for the rapid detection of wastes and the on-site environmental inspection.

Terahertz (THz) waves (0.1–10.0 THz) exist between the millimetre wave and the infrared band, which is significant in the biological sciences because of the complementary information that is available via traditional spectroscopic measurements on low-frequency bond vibrations, hydrogen bond stretching, and bond torsions in liquids and gases [12,13]. Combined with its nondestructive, accurate and rapid capabilities, and good penetrability [14,15], THz technology has many potential applications in materials [16,17], communications [18], biomedical detection [19], and other fields.

Here, taking pyrazinamide, benazepril, cefprozil, and bisphenol A as examples, we use THz waves to test their characteristic spectra and establish the concentration gradients. Additionally, combing with chemometrics, we quantitatively analyze their ratios in actual mixtures and compare this method with the Raman method. We expect our new method for the rapid qualitative and quantitative identification of industrial solid waste to be extended to other wastes.

## 2. Materials and Methods

### 2.1. Sample Preparation

Pyrazinamide (CAS: 98-96-4), benazepril (CAS: 86541-74-4), cefprozil (CAS: 92665-29-7), bisphenol A (CAS: 80-05-7), and high-density polyethylene (PE, CAS: 9002-88-4) were purchased from Sigma-Aldrich company (St. Louis, MO, USA), and all of them were powder samples. The PE powder was selected as a diluent to mix with the samples. Each pure substance was mixed with PE, ground into fine powder in an agate mortar, and pressed into a circular sheet with a thickness of ~1 mm and a diameter of ~13 mm under ~3 t pressure. The pressing time was set at 1 min. The masses of all the tablets were controlled to 90 mg.

For the pure substance test, we prepared 3 sheets for each group of substances, and each sheet contained 30 mg of substance and 60 mg of PE as diluent. For concentration analysis, each pure substance was mixed with PE in different proportions. The mass of the pure substance was 30 mg, 35 mg, 40 mg, 45 mg, and 50 mg, respectively. The mass of PE remained 60 mg unchanged, and the concentration of the sample was 33.3%, 36.8%, 40.0%, 42.9%, and 45.5%, respectively. When the mixing ratio of the tested sample to the background sample was 33.3–45.5%, the absorption peak of the sample could be distinguished obviously. If the mixing ratio was too large, it was easy to lead to absorption peak saturation, resulting in inaccurate results. For each group of different concentrations, we prepared 3 samples for parallel tests. For the mixtures test, we prepared four group mixtures with different weight ratios of pyrazinamide, bisphenol A, cefprozil, and benazepril, whose ratios were 6:7:5:7, 3:11:8:5, 4:12:5:5, and 5:9:7:4, respectively. We also prepared 3 sheets for each mixture, and each sheet contained 30 mg of mixture and 60 mg of PE. In addition, the pure PE powder was pressed into a sheet in the same way as the reference sample [20,21].

### 2.2. Instruments and Method

In this experiment, all the samples were tested by the Advan 7400 transmission terahertz time domain spectral system (THz—TDS), and each sample was tested four times. THz—TDS emitted wide-band terahertz wave with a frequency range of 0.5–4.0 THz and a spectral resolution of 7.9 GHz, and the signal-to-noise ratio (SNR) was 1000:1. Besides, all the spectra were averaged 512 times to ensure a high SNR. The entire system worked under an ambient temperature of 22 °C, and the humidity inside was ~3% to reduce the water absorption for the terahertz signal.

The THz—TDS system was used to obtain the time domain signal of sample, and the corresponding frequency domain signal was obtained by Fourier transform of the time domain signal:

$$E(\omega) = \int E(t)exp(-i\omega t)dt = M(\omega)exp[-i\phi(\omega)] \qquad (1)$$

where $E$ is the electric field, $M$ is the amplitude, $\phi$ is the phase, $t$ is time, and $\omega$ is the frequency.

According to Equation (1) and the tested signal in time domain, the reference frequency spectrum $E_r(\omega)$ and the sample frequency spectrum $E_s(\omega)$ can be calculated.

$$E_r(\omega) = M_r(\omega)exp[-i\phi_r(\omega)] \tag{2}$$

$$E_s(\omega) = M_s(\omega)exp[-i\phi_s(\omega)] \tag{3}$$

where $M_r$ and $M_s$ are the amplitude of the reference and sample, respectively. $\phi_r$ is the phase of reference and $\phi_s$ is the phase of sample. A transmission function $T(\omega)$ is obtained by Equations (2) and (3).

$$T(\omega) = \frac{E_s}{E_r} = \rho(\omega)exp[-i\varphi(\omega)] \tag{4}$$

where $\rho(\omega)$ is the amplitude ratio of the sample to the reference, $\varphi(\omega)$ is the phase difference between the sample and the reference.

Then, the absorbance of the samples $A(\omega)$ can be calculated by using the following equation [22,23].

$$I_r(\omega) = E_r(\omega) \times E_r(\omega)^* \tag{5}$$

$$I_s(\omega) = E_s(\omega) \times E_s(\omega)^* \tag{6}$$

$$A(\omega) = \frac{1}{d}ln\left(\frac{I_r(\omega)}{I_s(\omega)}\right) \tag{7}$$

$I(\omega)$ is the power spectrum. $I_r$ and $I_s$ are the power spectrum of the reference and sample, respectively.

The corresponding linear fitting function expression are labelled on each line. The expression for the function is:

$$y = ax + b \tag{8}$$

where $a$ and $b$ are constants, $x$ is the sample concentration, and $y$ is the amplitude or the area of absorption peak. $R^2$ is the correlation coefficient, which ranges from 0 to 1. The closer to 1, the higher the accuracy.

In the mixture analysis, we also used Lambert-Beer law:

$$A = \varepsilon l c \tag{9}$$

where $A$ is the absorbance, $\varepsilon$ is the molar absorption coefficient which is related to the nature of the absorption material and the wavelength of the incident light [24], $l$ is the thickness of the absorption layer, and $c$ is the concentration of the absorption material. We found that when the sample thickness was fixed, the absorbance of a specific sample was only positively correlated with the sample concentration [25].

Assuming that the mixture is composed of four substances, the absorbances of the four substances are $a$, $b$, $d$, $e$, and the absorbance of the mixture is $A$. The following equation can be obtained:

$$\begin{bmatrix} a_1 & b_1 & d_1 & e_1 \\ a_2 & b_2 & d_2 & e_2 \\ a_3 & b_3 & d_3 & e_3 \\ \vdots & \vdots & \vdots & \vdots \\ a_m & b_m & d_m & e_m \end{bmatrix} \cdot \begin{bmatrix} x_1 \\ x_2 \\ x_3 \\ x_4 \end{bmatrix} = \begin{bmatrix} A_1 \\ A_2 \\ A_3 \\ \vdots \\ A_m \end{bmatrix} \tag{10}$$

By solving Equation (10), we can get $x_1, x_2, x_3, x_4$, which is the concentration of each substance in the mixture.

## 3. Results and Discussions

### 3.1. THz Test Results

The THz spectra of four pure substances are shown in Figure 1, and the corresponding error bars have been labelled. The molecular structure of each substance is shown as the inset of each graph.

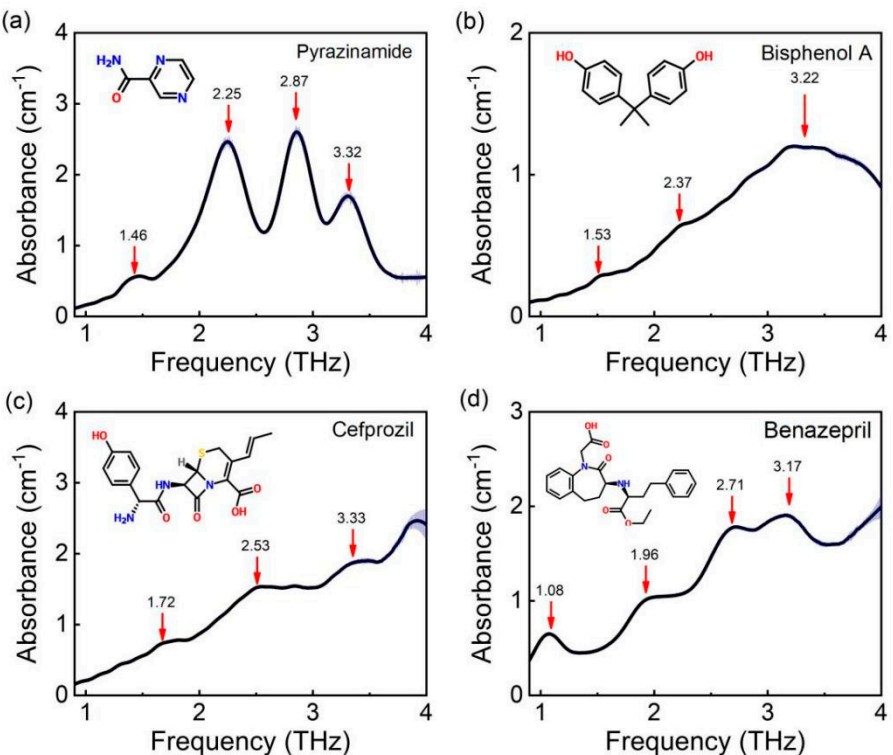

**Figure 1.** THz absorption spectra and molecular structures of four substances. (**a**) pyrazinamide, (**b**) bisphenol A, (**c**) cefprozil, and (**d**) benazepril. The corresponding error bars have been labelled (blue color).

As can be seen from Figure 1, four substances all have distinct and unique absorption peaks in the terahertz band. The characteristic peaks of pyrazinamide were 1.4, 2.2, 2.8, and 3.3 THz, respectively. The characteristic peaks of bisphenol A were 1.5, 2.2, 2.6, 3.1, and 3.5 THz, respectively. The characteristic peaks of cefprozil were 1.4, 1.7, 2.5, 3.3, 3.6, and 3.8 THz, and the characteristic peaks of benazepril were 1.1, 1.3, 1.9, 2.5, and 3.0 THz, respectively. The characteristic peaks of these substances were all quite different from each other, which is benefit for the later qualitative identification.

After the THz—TDS system was used to test and obtain the terahertz absorption spectra of the four substances, the refractive index and phase information of the four substances could also be obtained from the time-domain spectra, as shown in Figure 2.

It can be seen that the refractive index of the four substances shows an overall downward trend with the increase of frequency, and the phase shift of terahertz wave after passing through the samples of the four substances shows an overall upward trend with the increase of frequency, and the refractive index and phase shift are bisphenol A, cefprozil, pyrazinamide, and benazepril from high to low. In addition, comparing Figure 2a,b with Figure 1, it was found that near the frequency of the sample absorption peak, the refractive index of the sample and the phase shift of terahertz wave changed greatly, indicating the existence of abnormal dispersion.

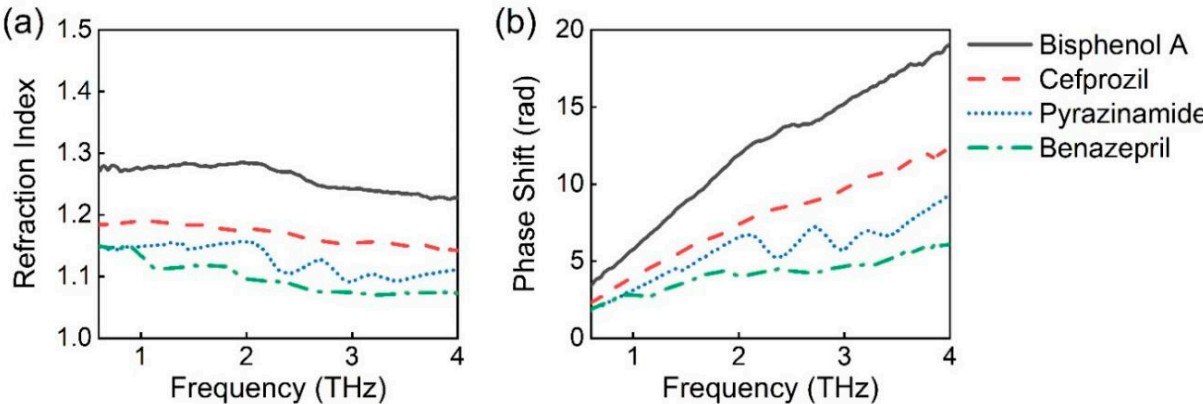

**Figure 2.** THz characteristic spectra of four samples. THz TDS system tests (**a**) Refractive index spectrum and (**b**) the phase spectrum of the four samples.

### 3.2. Concentration Analysis

After the characteristic spectra tests, we need to make the relationship between substance concentration and THz absorption peaks clear for the later quantitative analysis. Figure 3a–d presents the experimental absorption spectra of the samples under five different concentrations. For each substance, its spectra under different concentrations have the same waveform but different peak amplitude.

To further quantify the change of absorption peaks with the concentration, we extracted and fitted the corresponding absorption peak amplitudes of each sample under different concentrations. In addition, the absorption peak area as a function of the concentration was also calculated. Considering the four samples that were used in the experiment all have multiple absorption peaks, we selected the most obvious absorption peak for area analysis. Here, the characteristic peak amplitude value was determined by the highest absorbance point in the frequency range of the characteristic peak (2.25 THz for Pyrazinamide, 2.37 THz for Bisphenol A, 2.53 THz for Cefprozil, and 2.71 THz for Benazepril). Similarly, the range of the area under the peak was determined by the minimum point of the first derivative of the absorbance in the frequency range for the characteristic peak at the boundary (2.12–2.35 THz for Pyrazinamide, 2.26–2.45 THz for Bisphenol A, 2.43–2.60 THz for Cefprozil, and 2.61–2.80 THz for Benazepril). According to Bill Lambert's law and Equation (8), we linearly fit the above data, as shown in Figure 3e–l. The corresponding linear fitting function expression is marked on each line. In Figure 3, there are frequency shifts in some peaks. This is because the degree of interaction between the molecules varies with the concentration of the molecules. Some bonds are greatly influenced by molecular spacing and exhibit a gradual unidirectional frequency shift with increasing concentrations. Sometimes some unstable frequency offsets are due to jitter during system testing.

It can be seen from the Figure 3e–l that both the terahertz characteristic peak amplitude and the area under the absorption peaks have a linear relationship with the concentration. We can see that the values of $R^2_{THz}$ are between 90.3–99.8%. This high correlation coefficient proves that terahertz spectroscopy technology can perform high—precision quantitative detection of industrial solid waste concentration.

For comparison, we also tested these samples using the Raman method, which is also the lossless detection. The spectroscopy that we used was the Raman station 400 F, excitation wavelength was 532 nm, and the optional power was 250 mW. The results for the four pure substances and PE are shown in Figure 4.

It can be seen that pyrazinamide, bisphenol A, cefprozil, and PE all have distinct characteristic Raman peaks, while benazepril does not. Here, considering the range between 1063 cm$^{-1}$ and 1463 cm$^{-1}$ are easy disturbed by PE Raman peaks (gray shadow in Figure 5), we avoided this part in the later analysis. Except this range, the characteristic peaks of the samples can still be found: the Raman characteristic peaks for pyrazinamide were 808 and 1584 cm$^{-1}$; for bisphenol A was 835 cm$^{-1}$; and for cefprozil were 963, 2016, and 2059 cm$^{-1}$.

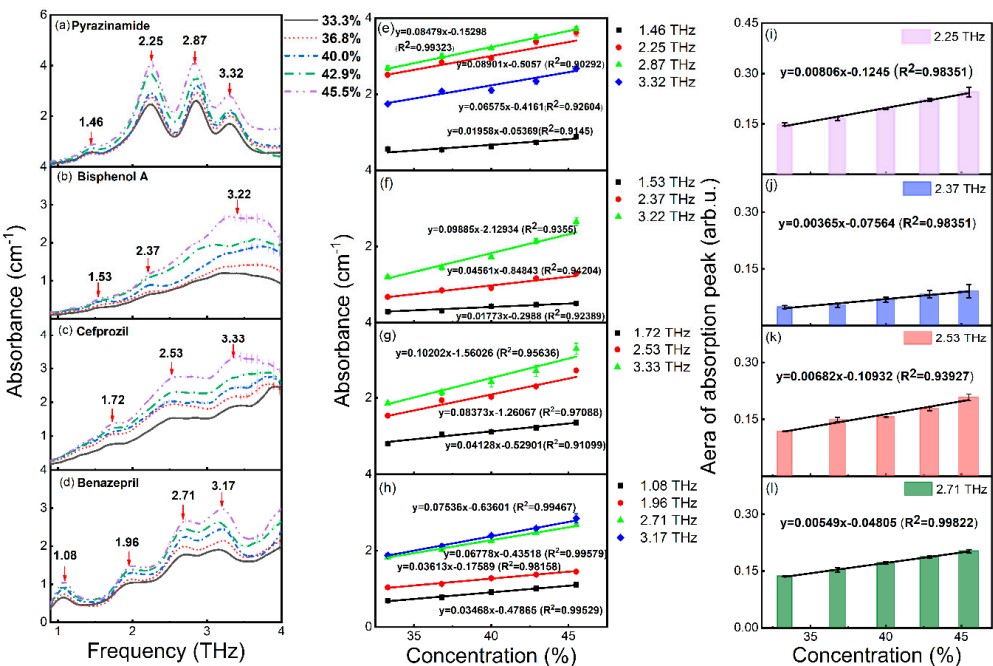

**Figure 3.** THz absorption spectra of (**a**) pyrazinamide, (**b**) bisphenol A, (**c**) cefprozil, and (**d**) benazepril under five different concentrations. The absorbance of absorption peaks for different concentrations of (**e**) pyrazinamide, (**f**) bisphenol A, (**g**) cefprozil, and (**h**) benazepril. The area under the absorption peaks of (**i**) pyrazinamide, (**j**) bisphenol A, (**k**) cefprozil, and (**l**) benazepril. The corresponding error bars have been labelled.

Next, we also did the linear fitting analysis for the substance concentration and the Raman characteristic peaks amplitudes. Figure 5 shows the Raman spectra of the four substances with different concentrations. From Figure 5a–d, it can be seen that the absorption intensity of samples increases with the increase of concentration. The peaks of PE cannot be seen from Figure 5c because the absorption intensity of cefprozil is much higher than that of PE. Similar to Figure 3, we used the same method to fit the intensity and area under the characteristic peak (see Figure 5e–j). Since benazepril has no characteristic absorption peak, we did not conduct further analysis. We can see that the range of $R^2_{Raman}$ only covers 11.7–86.9%, which is much lower than that of the THz method.

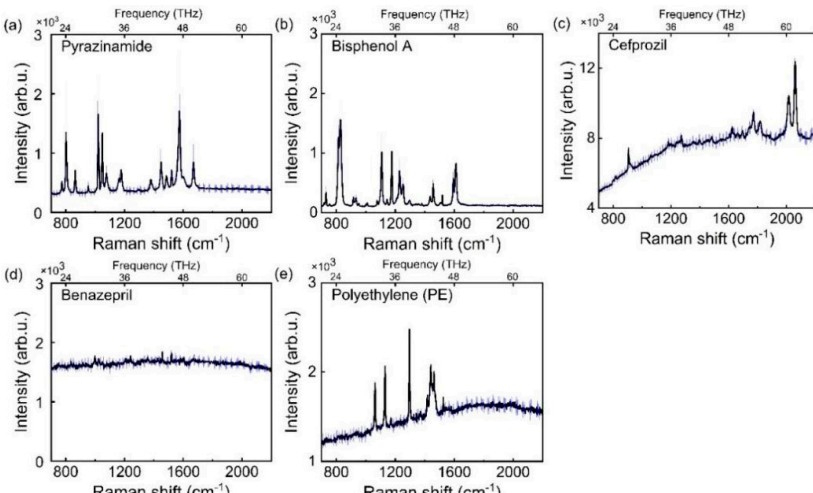

**Figure 4.** Raman spectra of four substances and PE (diluent): (**a**) pyrazinamide, (**b**) bisphenol A, (**c**) cefprozil, (**d**) benazepril, and (**e**) PE. The corresponding error bars have been labelled (blue color).

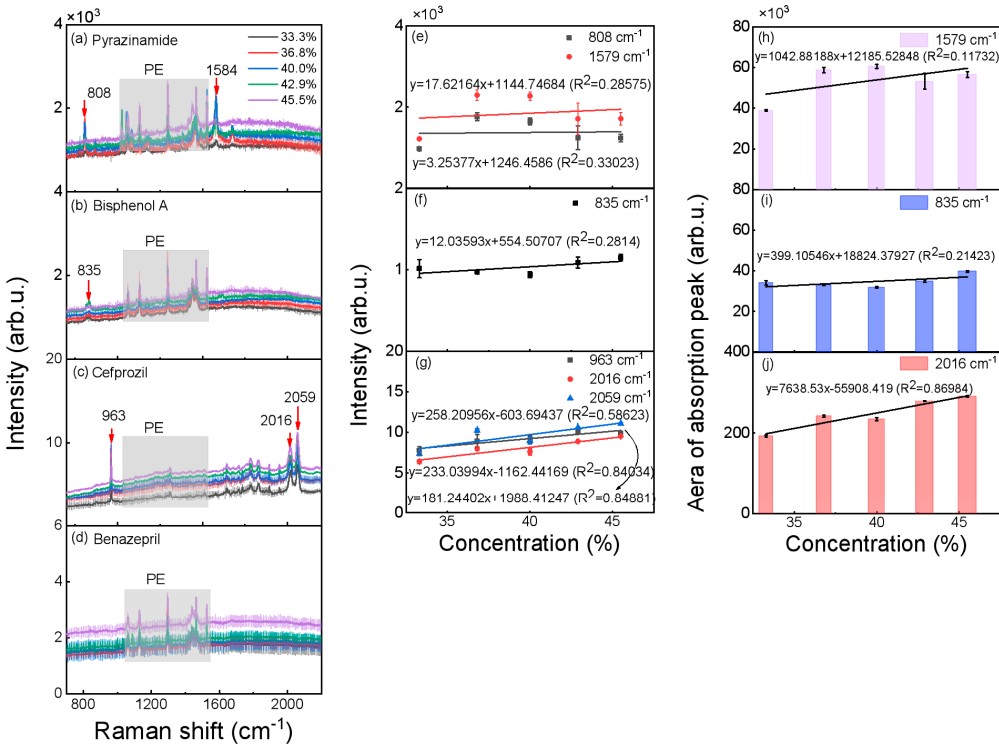

**Figure 5.** Raman spectra of (**a**) pyrazinamide, (**b**) bisphenol A, (**c**) cefprozil, and (**d**) benazepril under five different concentrations. The intensity of peaks for different concentrations of (**e**) pyrazinamide, (**f**) bisphenol A, and (**g**) cefprozil. The area under the peaks of (**h**) pyrazinamide, (**i**) bisphenol A, and (**j**) cefprozil. The corresponding error bar has been labelled.

### 3.3. Quantitative Analysis

Furthermore, we wanted to quantitatively analyze the specific proportion of harmful components in the industrial waste mixture. Here, we combined the terahertz spectral data of the mixture and the least square method to calculate the concentration of each substance in the mixture. To get this result, we also needed to measure the THz absorption spectrum of the different mixture samples. According to Lambert—Beer law (Equation (9)), and Equation (10) can be reduced to:

$$MX = A \tag{11}$$

By solving Equation (11), we can get $X$, which is the concentration of each substance in the mixture.

We prepared the mixture samples with the weight ratio of pyrazinamide, bisphenol A, cefprozil, and benazepril of 6:7:5:7, 3:11:8:5, 4:12:5:5, and 5:9:7:4, respectively, and repeated the THz spectral test. Here, the mixture ratios were random, we just used them for tests and model analysis.

Considering that there will be mass loss in the sample preparation, the quality difference can be obtained by weighing the mass of the sample after sheet pressing, and then the actual ratios are obtained by subtracting the difference value in proportion to the components in ideal ratio.

After testing the mixtures with different proportions of the four substances and combining their THz spectra of each pure sample, we use Equations (10) and (11) to fit the spectra of these mixtures, and the results are shown in Figure 6. In this figure, the left graphs show the experimental results of the mixtures with different mix—ratios, and the right graphs show the comparisons of the theoretically—predicted spectra based on the algorithm (red circle line) and the actual tested spectra (black square line). Table 1 shows the calculation results of the mix-ratios. It shows that for four mixtures with different mix—ratios, the errors between the actual results and the predicted calculation results are

less than 5%, which meet the certain precision requirements for the harmful substances detection in industrial wastes. It is proved that this method can be used to quantitatively analyze the components in the solid waste mixture.

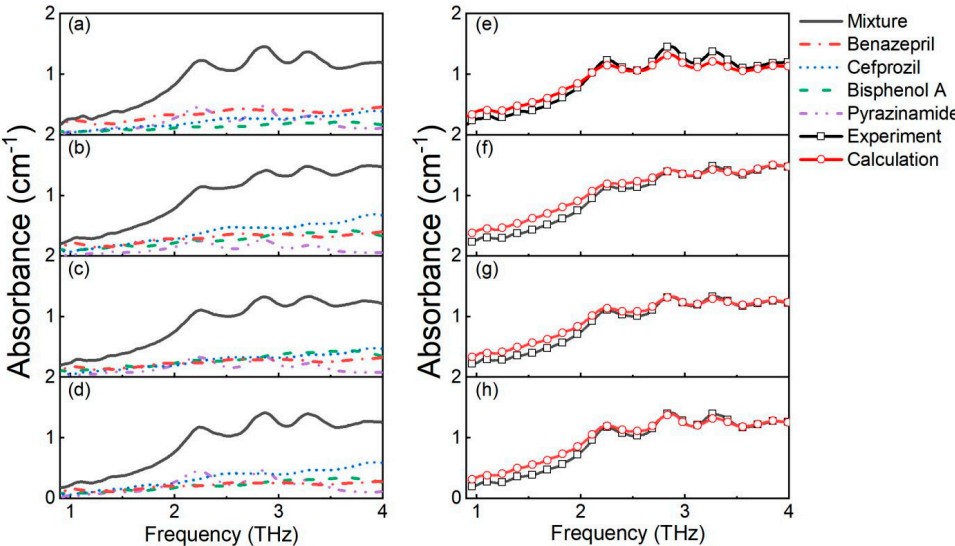

**Figure 6.** The THz spectra of the mixtures under different mix—ratio and the comparisons between the predicted and the actual spectra. The weight ratios of pyrazinamide, bisphenol A, cefprozil, and benazepril in mixture: (**a**,**e**) 6:7:5:7, (**b**,**f**) 3:11:8:5, (**c**,**g**) 4:12:5:5, and (**d**,**h**), 5:9:7:4 (**e**–**h**), respectively. The comparisons of the theoretically predicted spectra based on algorithm (red circle line) and the actual tested spectra (black square line).

**Table 1.** The quantitative analysis of the four substances in mixtures.

| Groups | Substance / Ratio | Pyrazinamide | Bisphenol A | Cefprozil | Benazepril |
|---|---|---|---|---|---|
| (a) | Ideal | 6.0 | 7.0 | 5.0 | 7.0 |
| | Actual | 5.6 | 6.5 | 4.6 | 6.5 |
| | Presumptive | 5.4 | 6.1 | 4.3 | 6.5 |
| | Relative error | 3.6% | 6.2% | 8.5% | 0% |
| | Average error | | 4.575% | | |
| (b) | Ideal | 3 | 11 | 8 | 5 |
| | Actual | 2.9 | 10.8 | 7.8 | 4.9 |
| | Presumptive | 2.9 | 11.1 | 7.3 | 5.1 |
| | Relative error | 0% | 2.8% | 6.4% | 4.1% |
| | Average error | | 3.325% | | |
| (c) | Ideal | 4 | 12 | 5 | 5 |
| | Actual | 3.8 | 11.4 | 4.8 | 4.8 |
| | Presumptive | 3.6 | 11.6 | 4.6 | 4.6 |
| | Relative error | 5.3% | 1.8% | 4.2% | 4.2% |
| | Average error | | 3.875% | | |
| (d) | Ideal | 5 | 9 | 7 | 4 |
| | Actual | 4.8 | 8.6 | 6.7 | 3.8 |
| | Presumptive | 4.8 | 9.0 | 6.2 | 3.9 |
| | Relative error | 0% | 4.7% | 7.5% | 2.6% |
| | Average error | | 3.7% | | |

## 4. Conclusions

In this paper, a detection method for harmful components in industrial solid waste is proposed. Taking pyrazinamide, benazepril, cefprozil, and bisphenol A as examples, the pure sample is detected by terahertz spectroscopy system. The linear relationship between the concentration and the peak area of terahertz spectrum is obtained, and the correlation coefficient is obtained. When the four harmful substances are mixed together, the specific harmful components in the mixture are identified by principal component analysis and least square method, and the content of each substance is quantitatively analyzed with an error of no more than 5%. The experimental results show that when terahertz technology is applied to the detection of industrial waste, it can accurately judge the types of substances, it can quantitatively analyze and detect the mixture, and the accuracy can reach 95%. This method is an efficient and accurate detection method in the field of industrial waste detection, which is worthy of further research and promotion.

**Author Contributions:** Q.W. (Qingfang Wang): Conceptualization, data curation, writing—original draft, investigation. Q.W. (Qichao Wang): Conceptualization, data curation, writing—original draft, investigation. Z.Y.: Conceptualization, supervision, writing—original draft, investigation. X.W.: Supervision, data curation, investigation, writing review and editing. Y.P.: Methodology, project administration, resources, writing review and editing, funding acquisition. All authors have read and agreed to the published version of the manuscript.

**Funding:** National Natural Science Foundation of China (81961138014, 61922059), Key projects of domestic scientific and technological cooperation in Shanghai (21015800200).

**Institutional Review Board Statement:** Not applicable.

**Informed Consent Statement:** Not applicable.

**Data Availability Statement:** Data underlying the results presented in this paper are not publicly available at this time but may be obtained from the authors upon reasonable request.

**Conflicts of Interest:** The authors declare that they have no known competing financial interest or personal relationships that could have appeared to influence the work reported in this paper.

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
