# Peer review of "Quantitative Analysis of Industrial Solid Waste Based on Terahertz Spectroscopy"

_photonics, doi:10.3390/photonics9030184_

Round 1

Reviewer 1 Report

The authors propose a method for the quantitative analysis of harmful components in industrial solid waste by using terahertz (THz) spectroscopy and compare it with Raman.  The author claimed to have much better accuracy in THz method.

However, readers might worry about the missing explanations on the underlying mechanisms.

  1. Polyethylene may be replaced with some other material for Raman measurement since it greatly affects the spectra of all the samples. Please comment on this and if possible try to repeat the experiment with some alternative to get some insight from Raman data.
  2. 2 quality is not up to the standard. It is blur and reader may have difficulty in reading. Author may redraw figures with better quality and used some graphical tool.
  3. Why concentration ratios of 33 to 45% are chosen. Higher concentration could have some interesting features.
  4. In Fig.2, there are frequency shift in some peaks. Can you please elaborate them in details about origin of this phenomenon?
  5. In Fig. 5, which algorithm is used to predict the spectra. Could you please provide the details about it.

Reviewer 2 Report

The manuscript of Qingfang Wang et al. is devoted to the quantitative THz spectroscopy of industrial solid waste. The experiment and its processing is of interest in THz and spectroscopy communities as well as important from the applied point of view. The manuscript could be accepted in Photonics after Authors comment the following issues:

  1. Keywords “mixture” and “quantitative analysis” are too general. Rephrase or replace them.
  2. Add the figure with your “row data”, i.e. with the examples of the measured waveforms.
  3. The bright absorption maxima are clearly seen for pyrazinamide in Fig only. 1. The maxima for other substances have a low-contrast, especially for cefprozil (Fig. 1c). The routine of the data processing allows one to reconstruct the phase [see Eq. (4)] and, therefore, the refractive index. Probably, the refractive index can provide better contrast. Please, add it to Fig. 1 in the case of better contrast.
  4. Why do the constant b depend on R^2? Is it a misprint?
  5. How did Authors calculate the area of absorption peak, shown in Fig. 2?
  6. Please, add the axis in THz into all panels of Fig. 3 for clear comparison with Fig. 1.

There are many comments here. They are not critical, but serious. So, I recommend Editor “Major revision”.

Reviewer 3 Report

In the manuscript, authors would propose a method for the quantitative analysis of 16 harmful components in industrial solid waste by using terahertz. The topic is interesting for the toxicity judgment of industrial solid waste. However, the paper mainly focus on powder samples, there is no discussion related to some solid waste.

Specific suggestions are as follows:

1, linear fitting function expression (formula 8) and formula 9 and 10 should be introduced in chapter two.

2, how to resolve the formula 10?

3, The conclusion is too simple, please give more descriptions.

4, In conclusion, authors proposed “an accuracy of more than 95%”, however, I fail to find the 95% in manuscript.

5, In part of material and method, please give some introduce about MCR. or there is no one understand the quantitatively analyse in chapter 3.2.

Round 2

Reviewer 1 Report

Authors have addressed almost all the point. Please reflect the following discussions in the manuscript for better readability. 

1-In Fig.2, there are frequency shift in some peaks. Please include your discussion regarding these shift in the manuscript.

2- selection of ratios should be discussed in the manuscript especially higher concentration effects.

Reviewer 2 Report

The authors have answered the questions and comments raised in my first report. I recommend publication of their manuscript in Photonics.

Author Response

Thank you very much for your valuable suggestions on this article, and especially for your recognition and support for the revised article.

Reviewer 3 Report

The author have coorected the problmes proposed by me. I agree to accept.

Author Response

(The authors gave the same response as above.)
